# “What’s This Bug?” Questions from the Public Provide Relevant Information on Species Distribution and Human–Insect Interactions

**DOI:** 10.3390/insects12100921

**Published:** 2021-10-09

**Authors:** André-Philippe Drapeau Picard, Marjolaine Giroux, Michel Saint-Germain, Maxim Larrivée

**Affiliations:** Montréal Insectarium—Space for Life, Montreal, QC H1X 2B2, Canada; marjolaine.giroux@montreal.ca (M.G.); michel.saint-germain@montreal.ca (M.S.-G.); maxim.larrivee@montreal.ca (M.L.)

**Keywords:** citizen science, human–insect interactions, insect conservation, public awareness, perception

## Abstract

**Simple Summary:**

Since its opening in 1990, the Montreal Insectarium has offered an entomological information service, allowing the public to send questions, photographs, and specimens for identification. All requests are answered by entomologists. Over the years, almost 14,000 requests have been received. We wanted to know which species have been seen over the years, which subjects were frequently asked about, and where requests came from. We analyzed the 4163 requests received in 2010–2011 and 2017–2018. Requests received during those four years came from 35 countries, and most of those requests came from Canada. Butterflies and moths were the most popular group. The five most frequent species were the eastern dobsonfly, the masked hunter, the giant water bug, the western conifer-seed bug, and the Japanese beetle. A comparison with the data from the citizen science platform iNaturalist shows that the EIS is a valuable tool to detect invasive species. Frequent subjects included school projects, entomophagy (eating insects), and wasp and bee nests.

**Abstract:**

In general, insects and arthropods polarizing: they either fascinate people, disgust people, or both, and they generate lots of questions. Museums are perceived as reliable sources of information and, as such, a go-to destination for the public to receive answers. Since its opening in 1990, the Montreal Insectarium has offered an entomological information service, allowing the public to send questions, photographs, and specimens for identification. All requests are answered by entomologists. Spatiotemporal variations in taxonomic, geographic, and thematic profiles of the 4163 requests received in 2010–2011 and 2017–2018 were analyzed. Requests came from 35 countries, and most of those requests came from Canada. The majority of requests were identification requests. Representing 25% of identification requests, the five most frequent species were the eastern dobsonfly *Corydalus cornutus*, the masked hunter *Reduvius personatus*, the giant water bug *Lethocerus americanus*, the western conifer-seed bug *Leptoglossus occidentalis,* and the Japanese beetle *Popillia japonica*. A comparison with the data from the citizen science platform iNaturalist shows that the EIS can be a valuable tool for invasive species detection. Frequent subjects included school projects, entomophagy (eating insects), and wasp and bee nests. Finally, we discuss the role of entomologists in providing scientific information but also in addressing common concerns regarding cohabitation with arthropods.

## 1. Introduction

Museums, besides conserving collections, are viewed as places of knowledge. As such, the public often seeks answers to their questions from museum personnel while visiting museums or through email. The Montreal Insectarium, hereafter the Insectarium, is one of North America’s most important insect museums. Open since 1990, it is part of Space for Life, a nature museum complex whose mission is to guide humans towards a fuller nature experience. Besides its exhibits and collections, the Insectarium supports the “Entomological Information Service” (EIS), providing the identification of specimens sent by mail or through pictures sent by emails as well as answering entomology-related questions. Such requests are received all year round and are addressed individually by entomologists.

The EIS was originally established to allow for a unidirectional flow of information from the museum’s experts to the general public. However, interest and participation from the public in scientific research and museology has grown in popularity and has gained recognition in recent decades [1,2]. The EIS is now seen as a bilateral exchange of information, where the public receives answers to their questions, and researchers are provided data relative to public interest and concerns regarding arthropods as well as observational data.

We summarize the requests made to the Insectarium’s EIS between 2009 and 2020 and present a detailed analysis of the requests received during two focus periods: 2010–2011 and 2017–2018. For those four years, we describe the thematic and taxonomic profiles of the requests received. We assess how the specimens and photographs received by the EIS compare to observations submitted to data from iNaturalist, a major citizen science platform. Finally, we discuss the role of entomologists in helping the public appreciate insects in daily life.

## 2. Materials and Methods

We summarized all of the requests received between 2009 and 2020, inclusively. We sampled four years, representing two focus periods: 2010–2011 and 2017–2018. Those years were assumed to be comparable, as the online submission form was the same, and the museum was open for the entirety of those focus periods. We also assumed that the two periods were far enough apart to detect eventual temporal changes.

We extracted taxonomic and geographic information from the requests received during the focus periods when possible. The EIS provides identification at least to the family level when requests include a specimen or a photograph of sufficient quality. Requests lacking such material were given a conservative taxonomic rank. For example, a request about general information on spiders was associated with the order Araneae since a lower taxonomy could not be determined, as no picture or specimen was provided.

Each request was associated with one or several categories, corresponding to themes observed by the museum’s entomologists while addressing the requests throughout years. The 14 categories are described in Table 1. Identification requests had their own category but could be associated with other categories as well.

We compared the number of identification requests received by the EIS to observations shared on the citizen science platform iNaturalist. We chose to compare the data for the five most abundant species in the requests submitted to the EIS overall, the five most abundant Lepidoptera species, and an invasive species recently introduced in Canada, the brown marmorated stink bug *Halyomorpha halys* (Stål, 1855) (Hemiptera, Pentatomidae). Data for those species from both sources were compared for the same geographic and temporal range, i.e., the province of Québec during the years 2010, 2011, 2017, and 2018.

## 3. Results

Between 2009 and 2020, the EIS received 13,973 requests, with a yearly average of 1164 (SD = 155.9). More than half of the requests were made between June and September, corresponding to the wild arthropod activity period at our latitude (Figure 1). While the monthly average over a year is 97 requests, it peaks at 230 for the month of July only. December has the lowest average, with 28.8 requests over the studied years. Interestingly, the 2020 summer months, during which sanitary measures were highly restrictive in many North American regions due to the COVID-19 pandemic, showed an exceptionally high number of requests.

### 3.1. Focus Periods (2010–2011 and 2017–2018)

During the focus periods (2010–2011 and 2017–2018), we received 4163 requests. The number of requests was similar between the two periods: 2121 were received in 2010 and 2011, while 2045 were received in 2017 and 2018. Most (70.5%) could be associated with a country. Those came from 35 countries, with the vast majority (96%) originating from Canada (Appendix A). Other countries were mostly French-speaking regions, e.g., France and Belgium, and touristic destinations, e.g., Mexico and the Caribbean. The number of countries was higher in 2010–2011 (32) than in 2017–2018 (19). Identification requests represented the majority (68.4%) of all of the requests. The other 31.6% was scattered among the categories shown in Table 1.

#### 3.1.1. Identification Requests

During 2010, 2011, 2017, and 2018, 2867 identification requests were submitted to the Entomological Information Service. Most (81.9%) identification requests could be identified at the family level. We were able to identify the genus for 56.7% of those requests, and the species in 46.3%. Overall, we documented 160 families, 283 genus, and 291 species. Lepidoptera was the most represented order, comprising 21.2% of identification requests, followed by the orders Hemiptera (17.6%) and Coleoptera (17.4%) (Figure 2). When looking at species diversity, identified lepidopterans and coleopterans included 115 and 88 species, respectively, while hemipterans included 40.

A total of 172 species were documented in 2010 and 2011 compared to 229 species in 2017–2018. Seventy-three species were common to both periods. The five most frequent species represented over 25% of all identification requests (Table 2). Those species are the eastern dobsonfly *Corydalus cornutus* (Linnaeus, 1758) (Megaloptera, Corydalidae), the masked hunter *Reduvius personatus* (Linnaeus, 1758) (Hemiptera, Reduviidae), the giant water bug *Lethocerus americanus* (Leidy, 1847) (Hemiptera, Belostomatidae), the western conifer-seed bug *Leptoglossus occidentalis* Heidemann, 1910 (Hemiptera, Coreidae), and the Japanese beetle *Popillia japonica*.

Two species were shared between the overall top five and focus periods. The Japanese beetle *P. japonica* Newman, 1838 (Coleoptera, Scarabaeidae) and the painted lady *Vanessa cardui* (Linnaeus, 1758) (Lepidoptera: Nymphalidae) replaced the cecropia moth *Halyophora cecropia* (Linnaeus, 1758) (Lepidoptera: Saturniidae) and the hummingbird clearwing *Hemaris thysbe* (Linnaeus, 1758) (Lepidoptera: Nymphalidae) between 2010–2011 and 2017–2018. The invasive *H. halys,* closely followed and was the sixth most frequent species in 2017–2018.

#### 3.1.2. General Information Requests

Besides identification requests, 1294 general entomological information requests were received during 2010, 2011, 2017, and 2018. They were distributed in 13 categories and corresponded to broad recurring themes (Table 1). The thematic profile did not vary between the 2010–2011 and 2017–2018 periods, apart from school projects, which were less frequent in 2017–2018. Requests about school projects were the most frequent category, representing 9.6% of general information requests, followed by entomophagy (7.4%), social wasp and bee nests (7.4%), and ants (7.1%). A significant number of requests (35.1%) could not be associated with a specific category.

The order Lepidoptera represented 23.4% of all general information requests, followed by Hymenoptera (20.7%) and Coleoptera (12.0%) (Figure 3). Requests about Lepidoptera were almost all about rearing monarchs *Danaus plexippus* (Linnaeus, 1758) (Lepidoptera: Nymphalidae) found in the wild and were associated with the category “Wild arthropod rearing”. Hymenoptera-related requests essentially consisted of concerns about wasp nests and ants near or inside houses, corresponding to categories “Social wasp and bee nests” and “Ants”.

#### 3.1.3. Comparison with Citizen Science Data

We compared the taxonomic profile of the identification requests made to the Entomological Information Service to the number of observations submitted to the citizen science platform iNaturalist for the same time period (2010, 2011, 2017, 2018) and the same geographic range (the province of Québec). We found that 7 out 11 species were better represented in the EIS requests than on iNaturalist for the spatiotemporal range we compared (Figure 4). Species occurrence was up to 6.7 times higher among EIS requests than they were on iNaturalist. Species that were represented particularly more in the EIS requests were the *C. cornutus*, *R. personatus*, *L. americanus,* and *H. cecropia*. There were 24 requests concerning *H. halys* compared to the 9 observations submitted on iNaturalist.

The EIS received many requests concerning invasive species. Some of those requests were used to track the spread of recently introduced species. There were 24 requests concerning *H. halys* compared to 9 observations submitted on iNaturalist. Requests about *H. halys* and *P. japonica* have been used to track the spread of those species as well as to track a parasitoid in Québec [3,4].

## 4. Discussion

Our study is one of a few to provide a quantitative analysis of requests from the public made to a science museum. During the four years that were investigated, the requests concerned almost 300 species, showing the diverse interests of the public. A few species represented 25% of all requests. The species that were most frequently observed by the people who reached out to the Insectarium’s Entomological Information Service were mostly large, associated with human habitations, or both. Our results also show that the requests to the EIS can be used to detect the presence of invasive species in urban areas [3,4]. Besides observations, the EIS received many requests about school projects, entomophagy, and concerns about hymenopterans near or inside houses. Documenting such trends helps museums to adapt their educational approach and to identify the need for complementary materials such as new web pages on various species.

In both identification and general information requests, the order Lepidoptera was the most represented, showing the interest that the public has for butterflies and moths. Attitudes towards lepidopterans are generally positive, and many of the requests we received about them concerned rearing, healing, or simply enthusiastically sharing sightings. Many requests concerning butterflies were related to school projects and could be linked to a monarch rearing program supported by the Insectarium. That program ended in 2016 due to concerns about monarch captive breeding [5], explaining the drop in school-related requests between the two focus periods. 

Variations in species composition between the different years reflected recent species introductions and peaks in the abundance of some species. The brown marmorated stink bug *H. halys* and the Japanese beetle *P. japonica* were introduced to North America and recently reached Canada [6,7]. While only a few requests were related to those species in 2010–2011, they were among the most abundant in 2017–2018. The painted lady *V. cardui* was the fourth most observed species in 2017–2018. This was mainly due to the climatic conditions that caused butterflies to settle around Montreal in high numbers during their fall migration in 2017. This phenomenon was also detected on citizen science platforms such as eButterfly [8].

The interest in entomophagy is also a sign of a positive attitude towards insects. Although eating insects is still a marginal practice in western countries, it has gained interest in recent years [9]. However, the number of requests related to entomophagy did not increase between the two focus periods. Many of those requests were also related to school projects. For years, the Insectarium has given its visitors the opportunity to taste insects. The museum is therefore seen as reliable source of information about entomophagy. 

Other requests were often associated with concerns. For example, requests about social hymenopteran control were frequent, reflecting concerns about human health and building integrity. Species that are often observed are easily noticeable either due to their size, their proximity to human habitations, or both. Such traits, along with a lack of knowledge in the species’ natural history, can generate both surprise and fear. For example, the giant water bug *L. americanus,* which is remarkable in size, is attracted to lights and is often found in swimming pools. Several people who contacted the EIS were surprised that such large insects could be found in Québec, with some of them even thinking they could have been introduced from tropical regions.

Although arthropods live all around us, their high diversity makes it difficult to know them all. The fear of the unknown as well as sensationalist media coverage results in insects and spiders being generally seen as threats to the health of humans, pets, and crops [10]. This is especially true in urbanized areas, which are associated with reduced insect knowledge [11]. Since the vast majority of insect and spider species are harmless to humans, their perceived threat is generally exaggerated. Therefore, entomologists play an educational as well as a calming role [12,13].

The vast majority of requests received by the EIS are identification requests associated with a photograph, a date, and a location. Thus, they can be considered observational data. We showed here that requests from the public can provide more information than a major citizen science platform such as iNaturalist, especially for invasive species in urban areas such as the brown marmorated stink bug *H. halys*. This could be explained by differences in the audiences who contact the Insectarium and iNaturalist. A person being intrigued by an insect found in the backyard is not necessarily interested in becoming involved in a community for naturalists and would likely prefer to contact the museum to respond to a punctual need. Citizen science has gained in popularity and recognition over recent years [2,14]. Traditionally, citizen science data have been collected through structured programs, but researchers are now using various sources such as social media [15].

## 5. Conclusions

While insects and other arthropods provide important ecosystem services, many people have exaggerated concerns about them. Given the diversity of invertebrates, it can be difficult for the public to put a name to what they see and to receive the information they need by themselves. Therefore, entomologists can act as agents between the public and insect knowledge to foster cohabitation. On the other hand, questions from the public inform entomologists about what people see and care about insect-wise. They provide information that is comparable to citizen science platforms for some species. As such, they can be used for species detection, such as invasive species. In other words, learning about insects creates knowledge about humans, and vice versa.

## Figures and Tables

**Figure 1 insects-12-00921-f001:**
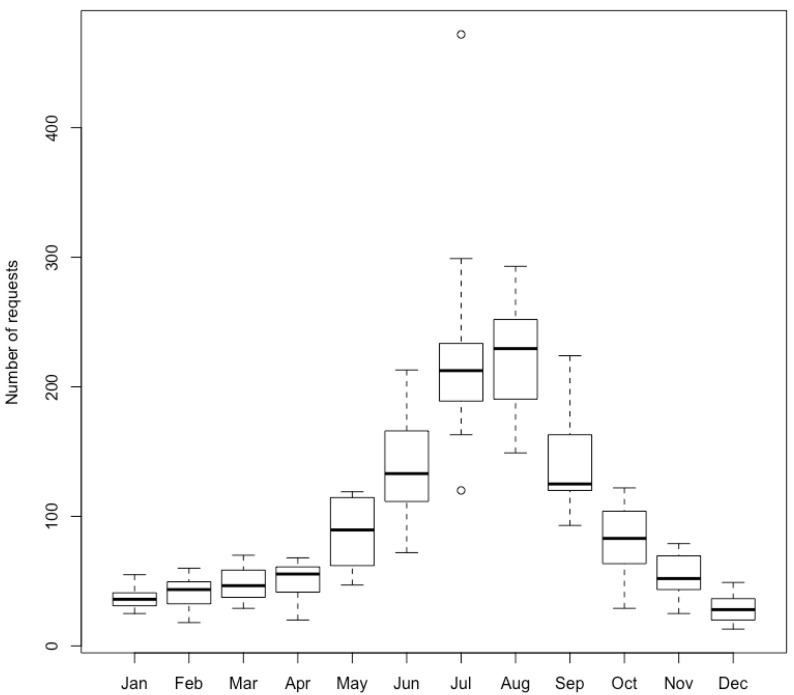
Sum of requests received monthly by the Montreal Insectarium’s Entomological Information Service between 2009 and 2020, inclusively.

**Figure 2 insects-12-00921-f002:**
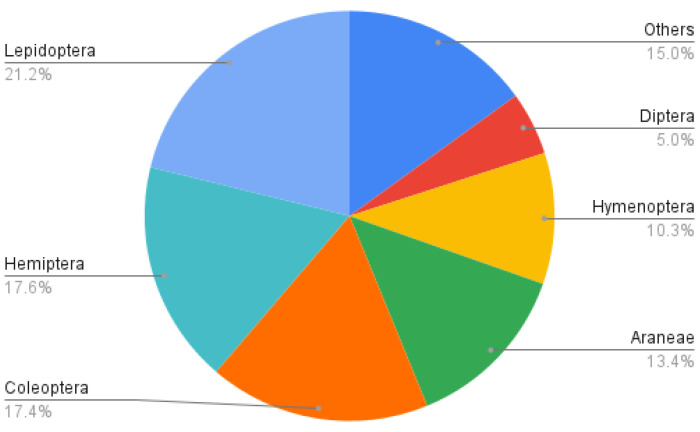
Order-level taxonomic distribution of identification requests received by the Montreal Insectarium’s Entomological Information Service in 2010, 2011, 2017, and 2018.

**Figure 3 insects-12-00921-f003:**
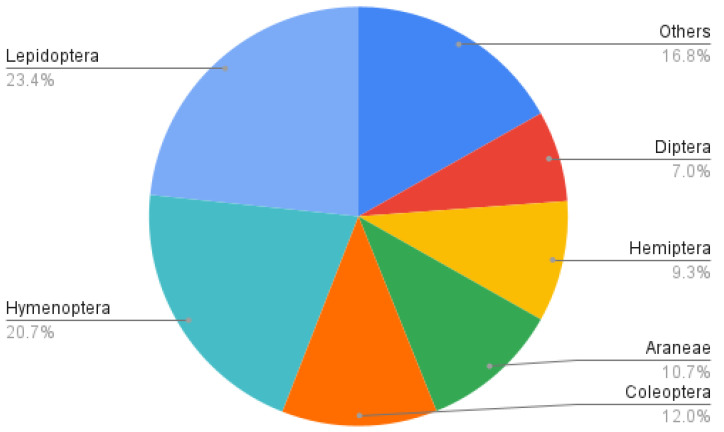
Order-level taxonomic distribution of general information requests received by the Montreal Insectarium’s Entomological Information Service in 2010, 2011, 2017 and 2018.

**Figure 4 insects-12-00921-f004:**
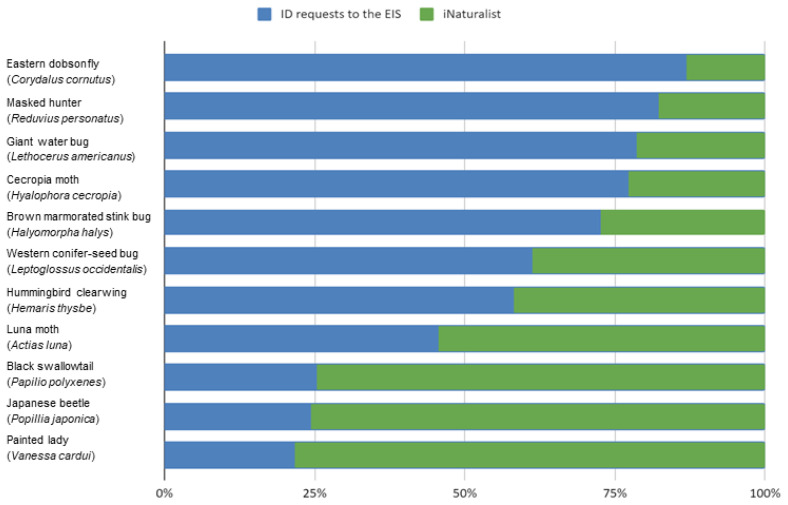
Comparison of the relative number of requests submitted to the Insectarium’s Entomological Information Service with observations submitted to iNaturalist for the province of Québec during 2010, 2011, 2017, and 2018.

**Table 1 insects-12-00921-t001:** Thematic distribution of requests received by the Montreal Insectarium’s Entomological Information Service in 2010, 2011, 2017, and 2018. A given request can be associated with more than one category.

Category	Description	Number of Requests Total (%)
School projects	Teachers or students reaching out about their projects	138 (9.72)
Entomophagy	Requests related to willful insect consumption by humans	106 (7.46)
Social wasp and bee nests	Requests about concerns or how to deal with social wasps and/or bee nests, generally near houses or in public parks	106 (7.46)
Ants	Requests about concerns or how to deal with ants, generally inside houses	102 (7.18)
Wild arthropod rearing	Requests about how to rear arthropods found in the wild, e.g., caterpillars	90 (6.33)
Buying arthropods	Requests about where to buy live arthropods, e.g., for biocontrol	81 (5.70%)
Insect collecting, mounting and curating	Requests about where to find entomological gear, how to kill and preserve specimens, etc.	51 (3.59)
Non-arthropods	Requests about animals other than arthropods, mostly invertebrates such as molluscs and annelids	29 (2.04)
Arthropods found in commercial food	Reports of incidental observations of arthropods on/in food from the store, e.g., spiders found on grapes	27 (1.90)
Terminology and translation	Requests about entomological terms or insect common names, especially from the media or other museums	20 (1.41)
Arthropod behavior	Questions about specific or general behaviors such as insect flight, migration, or overwintering.	20 (1.41)
Ekbom’s-syndrome-like	Requests from people possibly affected by mental issues concerned about insects invading their body or home	15 (1.06)
Other general entomological information	Requests related to entomology, but not matching other categories	499 (35.14)
Other	Requests not related to entomology	128 (9.01)

**Table 2 insects-12-00921-t002:** Comparison of the five most frequent species for the two focus periods and for all years, based on the number of requests (N) received by the Montreal Insectarium’s Entomological Information Service.

All Years		2010–2011		2017–2018	
Species Name	N	Species Name	N	Species Name	N
Eastern dobsonfly (*Corydalus cornutus*)	102	Eastern dobsonfly (*Corydalus cornutus*)	54	Japanese beetle (*Popillia japonica*)	59
Masked hunter (*Reduvius personatus*)	79	Giant water bug (*Lethocerus americanus*)	52	Eastern dobsonfly (*Corydalus cornutus*)	48
Giant water bug (*Lethocerus americanus*)	66	Masked hunter (*Reduvius personatus*)	45	Masked hunter (*Reduvius personatus*)	34
Japanese beetle (*Popillia japonica*)	63	Cecropia moth (*Halyophora cecropia*)	31	Painted lady (*Vanessa cardui*)	31
Western conifer-seed bug (*Leptoglossus occidentalis*)	63	Hummingbird clearwing (*Hemaris thysbe*)	28	Western conifer-seed bug (*Leptoglossus occidentalis*)	30

## Data Availability

The data presented in this study are openly available on the Open Science Framework at https://osf.io/fdk9u (accessed on 2 October 2021).

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
