# Peer review of "“What’s This Bug?” Questions from the Public Provide Relevant Information on Species Distribution and Human–Insect Interactions"

_insects, 2021, doi:10.3390/insects12100921_

Round 1

Reviewer 1 Report

This paper summarizes data on requests for information and/or identifications of insects and related arthropods submitted to the Montreal Insectarium’s Entomological Information Service during the period 2009 to 2020. These summaries provide insight into the species and higher groups that are most likely to capture the attention of the general public and also indicate which insect groups tend to cause the greatest anxiety. The paper is well-written and certainly publishable, but I would like to have seen the authors go a bit further with their analysis. For example, it is not clear why they chose to include only data for two 2-year periods, 2010-2011 and 2017-2018, in their analysis rather than all of the data. Also, I was expecting to see more comparisons between the earlier and later dataset to highlight possible changes in the overall pattern of requests, species/groups emphasized, etc., over time. Instead they apparently just lumped the data for these 4 years together to produce the summaries in Table 1 and Figs 2-4.  The Results do mention that the number of countries from which requests originated, and the number of species for which information was requested, differed between the two periods; they also mention that requests for information on the recently introduced brown marmorated stinkbug began only after the earlier period. However, it should be possible to provide more details on changes over time in requests for information on other commonly observed species or groups and using the entire dataset rather than just two 2-year snapshots would be more informative. Presumably the authors didn’t have time to do this more detailed analysis, which would be understandable, but this should be explained more clearly in the Methods. Still, it may not be too difficult to examine a few additional species or groups in detail and trace trends in observations of these groups over the entire period. The most unexpected result for me was that requests related to entomophagy were tied for second place in frequency (after school projects). Could the authors provide further comments on why this might be, whether any particular species were commonly mentioned, and whether there is any temporal trend in the data related to this category (i.e., was there a trend toward increasing requests over time)?

Author Response

While we do have and present the number of requests received between 2009 and 2020, we only went through requests from the two focus periods to extract taxonomic and geographic information. The process was time-consuming so we chose to use those four years as samples. We think to changes in wording in the Methods section is now clearer on that aspect.

Details were added for comparison between the two focus periods in terms of species composition. Table 2 and a paragraph were added to the Results section to compare differences in most observed species between the two focus periods, and overall. We further discuss those differences in a new paragraph in the Discussion section.

We did not find marked differences in subjects between the two focus periods, except for school-related requests. Therefore, we did not include much detail about thematic changes. However, we added details about entomophagy-related requests in the Discussion sections. Specifically, we added a paragraph explaining the possible causes of the relatively high number of such requests.

Thank you for reviewing our manuscript!

Reviewer 2 Report

The comparison with iNaturalist is interesting, and could appear as one of the noteworthy conclusions, mentioned in the abstract.  In contrast, the claim that these data can be used to "track the spread of introduced species" is not clearly supported by the results.  Introduced species were indeed  reported, but the paper does not provide documentation of "tracking," i.e. monitoring range expansion over time.  In general, I think that both the abstract and the conclusions section should reflect the findings that are summarized in the figures and tables.  As it stands, they are both rather vague.

Otherwise, the paper is clearly written, nicely presented, and an interesting contribution.  

Author Response

Details were added to the Simple Summary, Abstract, and Conclusions sections so they better reflect the results. The wording of the sentences regarding species tracking was changed. We now use “detection” instead of “tracking”. Thank you for reviewing our manuscript and for your good words!